# Deep Graph Mating

**Yongcheng Jing**[1]  **Seok-Hee Hong**[1]  **Dacheng Tao**[2]
[1]University of Sydney  [2]Nanyang Technological University
{yongcheng.jing,seokhee.hong}@sydney.edu.au, dacheng.tao@ntu.edu.sg

## Abstract

In this paper, we introduce the first learning-free model reuse task within the non-Euclidean domain, termed as *Deep Graph Mating* (GRAMA). We strive to create a child Graph Neural Network (GNN) that integrates knowledge from pre-trained parent models without requiring re-training, fine-tuning, or annotated labels. To this end, we begin by investigating the permutation invariance property of GNNs, which leads us to develop two vanilla approaches for GRAMA: *Vanilla Parameter Interpolation (VPI)* and *Vanilla Alignment Prior to Interpolation (VAPI)*, both employing topology-independent interpolation in the parameter space. However, neither approach has achieved the anticipated results. Through theoretical analysis of VPI and VAPI, we identify critical challenges unique to GRAMA, including increased sensitivity to parameter misalignment and further the inherent topology-dependent complexities. Motivated by these findings, we propose the *Dual-Message Coordination and Calibration (DuMCC)* methodology, comprising the *Parent Message Coordination (PMC)* scheme to optimise the permutation matrices for parameter interpolation by coordinating aggregated messages, and the *Child Message Calibration (CMC)* scheme to mitigate over-smoothing identified in PMC by calibrating the message statistics within child GNNs. Experiments across diverse domains, including node and graph property prediction, 3D object recognition, and large-scale semantic parsing, demonstrate that the proposed DuMCC effectively enables training-free knowledge transfer, yielding results on par with those of pre-trained models.

## 1 Introduction

The remarkable progress made in deep neural networks has resulted in a growing number of pre-trained models being made publicly available for the purpose of performance reproducibility and further development [64, 12, 27, 23, 61, 24]. As such, there is a mounting interest in the community on the reusability of existing pre-trained neural networks for the sake of strengthening performance, reducing model size, or alleviating training efforts, with abundant inspiring works proposed [16, 42, 46, 40, 47, 45, 63, 67]. Despite the growing interests in model reuse, current research endeavors have predominantly focused on the Euclidean domain, which are specifically designed to handle image data with regular grid structures [39, 64, 34, 12, 51, 14, 22].

On the other hand, the study of reusing pre-trained *Graph Neural Networks (GNNs)* to tackle non-Euclidean and irregular graph data is still in its early stage and remains limited in scope. Almost all existing research on GNN reuse is established upon non-Euclidean *Knowledge Distillation (KD)* pioneered by Yang et al. [60], where a favourable student GNN is learned from a single pre-trained teacher [11, 49, 9, 66, 29]. Subsequent research has extended the scope of Yang et al. [60] from a single-teacher to a multi-teacher context, introducing the novel task of non-Euclidean *Knowledge Amalgamation (KA)* [25, 15, 36]. However, all these existing approaches to GNN reuse necessitate resource-intensive *re-training* of the student model, imposing substantial computational burdens and

| Graph-Centric Model Reuse Tasks | Multi-model Reuse | Annotation Free | Training/Fine-tuning Free |
|---|---|---|---|
| Knowledge Distillation [60, 11, 49, 9, 66, 29] | × | × | × |
| Knowledge Amalgamation [25, 15, 36] | ✓ | ✓ | × |
| Deep Graph Mating (GRAMA) | ✓ | ✓ | ✓ |

Table 1: Comparison of various model reuse tasks in the non-Euclidean domain, tailored for GNNs.

memory costs. This challenge from *re-training* is particularly pronounced when dealing with large models and large-scale graphs [31, 7, 37, 20, 8, 26, 17].

In this paper, we strive to push the boundaries of resource-efficient GNN reuse by introducing the first *training-free* model reuse task in the non-Euclidean domain, termed as *Deep Graph Mating* (GRAMA). Our objective is to derive a child GNN, *without re-training* or *fine-tuning*, from pre-trained parent GNNs, each possessing unique knowledge from different datasets, all while operating without access to human-annotated labels—a common constraint in using publicly available models. The child model born from GRAMA is expected to seamlessly integrate the diverse expertise of its parent models in a completely *learning-free* manner. A comparative analysis of GRAMA with existing non-Euclidean model reuse approaches is presented in Tab. 1. As a pilot study of this novel task, this paper focuses on homogeneous GRAMA scenarios with identical parent architectures, and sets the stage for future investigations into the more complex cross-architecture heterogeneous GRAMA.

To achieve the ambitious goals of GRAMA, we first investigate the permutation invariance property of GNNs to establish correspondence between neurons in pre-trained models. This investigation guides the development of two vanilla methods: *Vanilla Parameter Interpolation (VPI)* and *Vanilla Alignment Prior to Interpolation (VAPI)*. VPI involves straightforward linear interpolation of parent GNN parameters, while VAPI incorporates data-independent parameter matching before interpolation. However, the performance of both VPI and VAPI has not been as promising as expected. By analysing the mechanisms underlying these methods, we have identified and theoretically demonstrated unique challenges associated with GRAMA, including increased susceptibility to parameter misalignment and topology-dependent complexity in GNNs. These findings underscore the necessity for tailored methods that are specifically adapted to GRAMA.

To this end, we introduce the *Dual-Message Coordination and Calibration (DuMCC)* methodology, specifically tailored for GRAMA to incorporate the topological characteristics of graphs. DuMCC consists of two distinct schemes: *Parent Message Coordination (PMC)* and *Child Message Calibration (CMC)*. PMC seeks to identify optimal permutation matrices for parameter matching in a topology-aware manner, by coordinating aggregated messages from parent GNNs. Although PMC shows promising results, our empirical and theoretical analyses indicate that the child GNN derived from this coordination is more prone to over-smoothing. To mitigate this issue, we propose the CMC scheme, which calibrates the message statistics of the child GNN using a specialised, learning-free message normalisation (LFNorm) layer, drawing on the statistics of the parent GNNs. Together, these two schemes contribute to a training-free and label-free GRAMA process, enabling the derivation of child GNNs that effectively embody the knowledge from their pre-trained parent models.

In summary, our contribution is a novel non-Euclidean model reuse paradigm that allows for the creation of a student GNN, which integrates the capabilities of pre-trained parent GNNs without requiring human-annotated labels or re-training. We evaluated our proposed approach on seven benchmarks across various tasks, including node/graph classification, object recognition, and large-scale indoor semantic segmentation, and across five different GNN architectures: Graph Convolutional Network (GCN) [30], GraphSAGE [13], Graph Attention Network (GAT) [48], Graph Isomorphism Networks (GIN) [56], and Dynamic Graph CNN (DGCNN) [53]. Experimental results demonstrate that the generated child GNN is competent to handle the pre-trained tasks of parent models. We also present a discussion of the limitations, highlighting potential future research directions that can be explored based on GRAMA.

## 2   Related Work

This section briefly reviews topics relevant to GRAMA, including existing GNN reuse techniques and model merging, which, while conceptually akin to GRAMA, is confined to the Euclidean domain. Extended related work is provided in Appendix A.

**Non-Euclidean Model Reuse.** Recent advancements in graph-centric intelligence have led to the widespread availability of pre-trained models for reproducibility. Despite their availability, the exploration of their reuse for downstream tasks, particularly in the non-Euclidean domain, is still in its infancy. Existing research primarily revolves around knowledge distillation (KD) [60, 9, 66, 11, 49, 29] and knowledge amalgamation (KA) [25, 15, 36], aimed at single-GNN and multi-GNN reusing, respectively. In particular, the foundational KD research [60] introduced a method specifically designed for GNNs that conserves local structural integrity. The scheme was later expanded by KA [25] to adapt to multi-teacher settings without the need for labels, thereby enhancing GNN reuse capabilities. Extending these concepts, this paper launches GRAMA, a new GNN reuse paradigm that further boosts resource efficiency by obviating the need for re-training.

**Model Merging.** Recent years have seen a surge in interest in merging the weights of CNNs or transformers into a single model [54, 43, 1, 19, 44, 57, 33, 59, 35, 62, 38, 2, 21]. Model merging is typically categorised into two types: fine-tuned model merging and variably initialised model merging [33, 44]. Merging models fine-tuned from the same initialisation is generally straightforward as these models often reside within the same error basin, allowing for simple weight interpolation [54]. Conversely, merging models from different initialisations is more challenging due to the randomness in network channels and components [33, 43, 1]. A key issue is aligning neurons between models to establish correspondences before weight interpolation. To address this issue, Git Re-basin [1] proposes to minimise the $L_2$ distance between weight vectors. Singh and Jaggi [43] propose an Optimal Transport Fusion (OTFusion) method that uses the Wasserstein distance to align weight matrices prior to performing parameter fusion. The subsequent work [19] extends the application of OTFusion to Transformer-based architectures, aiming to enhance efficiency and performance through fusion. Also, Liu et al. [35] approach the challenging task of model fusion as a graph matching problem, incorporating second-order parameter similarities for improved fusion performance.

Addressing the fusion of pre-trained models trained on disparate tasks, Stoica et al. [44] develop ZipIt!, a novel method that utilises a "zip" operation for layer-wise fusion based on feature redundancy, creating a versatile multi-task model without additional training. More recently, Xu et al. [57] present the Merging under Dual-Space Constraints (MuDSC) framework. This approach optimises permutation matrices by mitigating inconsistencies in unit matching across both weight and activation spaces, targeting effective model fusion in multi-task scenarios. Furthermore, Jordan et al. [28] introduce REPAIR, a method that tackles the issue of variance collapse by rescaling the hidden units in the interpolated network to restore the statistical properties of the original networks.

However, model merging has not yet been explored within the context of GNNs. This paper introduces GRAMA, the first formulation for weight space model merging tailored for the non-Euclidean domain. It represents the initial investigation into merging GNNs, addressing the unique challenges that arise with graph tasks.

## 3 Motivation and Problem Definition

In this section, we begin by introducing *Knowledge Amalgamation (KA)*, the sole existing task in multi-GNN reuse, exploring its inherent limitations, and present our novel GRAMA paradigm for resource-efficient multi-model reuse with practical applications in real-world scenarios.

To the best of our knowledge, KA represents the only multi-model reuse task in the non-Euclidean domain. KA is defined in the literature as follows: its objective is to learn a single, compact student GNN that integrates the diverse expertise of pre-trained teacher GNNs, without accessing human annotations [25]. Despite the promising performance, KA is inherently limited by its resource-intensive nature, requiring the re-training of a student GNN to amalgamate knowledge from existing GNNs. Furthermore, while KA ostensibly operates without ground-truth labels, it instead relies on soft labels generated by teachers, making it susceptible to the misclassification errors of teachers.

To mitigate these constraints of KA, this paper proposes:

**Task 3.1** (Deep Graph Mating). *Deep Graph Mating (GRAMA) is a fully learning-free model reuse task where a child GNN is derived from pre-trained parent GNNs without re-training or fine-tuning, integrating their expertise without requiring human-annotated labels.*

In aggregate, GRAMA advances beyond KA by eliminating the need for any training or label dependency, paving the way for more widespread and versatile model reuse applications. Given the

novelty and complexity of GRAMA, our initial investigation in this paper is confined to scenarios where pre-trained GNNs possess identical architectures yet are trained on separate datasets, termed as homogenous GRAMA. We reserve the exploration of more challenging heterogeneous GRAMA scenarios, where pre-trained parent models either have diverse architectures or are designed for different domain tasks, as a topic for future studies, potentially incorporating solutions like partial GRAMA inspired by the work of Stoica et al. [44].

The applications of homogenous GRAMA are especially crucial in contexts where full data access for training is restricted due to privacy concerns and regulatory requirements. This is common in sectors like healthcare or retail, where organisations operate across different regions, each gathering data that cannot be centrally aggregated due to local privacy regulations. The proposed homogenous GRAMA paradigm enables the seamless integration of knowledge from these isolated datasets, thereby safeguarding against the risks of privacy violations and the disclosure of sensitive data.

## 4 Vanilla Methodologies and Challenge Pre-analysis

### 4.1 Two Vanilla GRAMA Methods

**Vanilla Parameter Interpolation (VPI).** To achieve the ambitious goal of GRAMA outlined in Sect. 3, the initial naïve approach employs vanilla weight averaging [50, 54]. This method involves a straightforward linear interpolation of weights $W_a$ and $W_b$ from two pre-trained GNNs: $W^{(\ell)} = \alpha W_a^{(\ell)} + (1 - \alpha) W_b^{(\ell)}$, where $\alpha$ represents the interpolation weight and $W^{(\ell)}$ denotes the network weights at layer $\ell$.

However, the vanilla averaging approach requires the pre-trained models to share a portion of their training trajectory and remain sufficiently close in the parameter space [33], typically achieved by fine-tuning from the same initial model. This is not applicable in our GRAMA context, where parent GNNs are trained on distinct datasets. This mismatch leads to empirically poor performance for GRAMA, as observed in our experiments.

**Vanilla Alignment Prior to Interpolation (VAPI).** To address this issue in vanilla interpolation, previous research in the Euclidean domain, based on the conjecture of the permutation invariance property of typical neural networks[1], proposes aligning the neurons between pre-trained models by permuting parameter matrices before performing linear interpolation [1]. The alignment and interpolation process can be formulated as:

$$W^{(\ell)} = \alpha W_a^{(\ell)} + (1 - \alpha) P^{(\ell)} W_b^{(\ell)} (P^{(\ell-1)})^T, \quad P^{(\ell)} \in \mathbf{P}^*, \tag{1}$$

where $\mathbf{P}^* = \left[ P^{(\ell)} \right]_{\ell \in [L]}$ represents the set of all permutation matrices $P^{(\ell)}$ for each layer $\ell$ of the GNN. Here, $[L]$ refers to the set of indices corresponding to all layers in the GNN.

However, to apply Eq. 1 to GNN-based GRAMA, it is essential to first discuss whether the permutation invariance property extends to GNNs. This property has been extensively studied in existing literature for various architectures [10, 5, 43, 1, 33, 19], including multi-layer perceptrons (MLPs), convolutional neural networks (CNNs), and Transformers. Supported by these prior studies, and given that GNNs are fundamentally built upon MLPs [58, 30], we propose:

**Conjecture 4.1** (Permutation Invariance in GNNs). *Permutation invariance for parameters in GNNs exists if and only if there exists a set of permutation matrices $P^{(\ell)}$ for each layer $\ell$ such that applying these permutations to the parameters does not alter the outcome of graph-based learning task, regardless of the structure of the adjacency matrix.*

A key subsequent issue involves searching the optimal permutation matrix $\mathbf{P}^*$ for GNNs. One possible data-independent solution is to minimise the $L_2$ distance between the weight vectors of the pre-trained models by solving a sum of bilinear assignments problem, similar to weight matching techniques described in [1]. This method, which does not consider data distributions, could be adapted as a baseline *vanilla* method for our GRAMA task in the non-Euclidean domain, which is evaluated in our experiments.

---

[1]Neurons in each layer of neural networks can be permuted without altering network functionality [10, 1].

## 4.2 Challenges Towards GRAMA

However, we empirically observed that even the *Vanilla Alignment Prior to Interpolation* method yielded unfavourable results for our GRAMA task. To elucidate the underlying cause of this phenomenon, we theoretically demonstrated that GNNs typically exhibit greater sensitivity to parameter mismatches than neural networks in the Euclidean domain:

**Lemma 4.1** (Amplified Sensitivity of GNNs to Parameter Misalignment)**.** *GNNs exhibit greater sensitivity to mismatches in parameter alignment compared to CNNs, amplified by the degree of connectivity and heterogeneity of the node features in the graph topology.*

A complete theoretical proof of Lemma 4.1 is provided in Appendix G.1. Here, we present only the final formulation of the approximated output changes resulting from weight perturbations due to mismatching, based on Taylor series approximation:

$$\Delta F_i \approx \sigma' \left( \sum_{j \in \mathcal{N}(i)} W \cdot X_j \right) \cdot \sum_{j \in \mathcal{N}(i)} \epsilon \cdot X_j, \tag{2}$$

where $\Delta F_i$ refers to the change in output at node $i$ due to the perturbations $\epsilon$ in the weights $W$, and $\sigma'$ represents the derivative of the activation function. $X_j$ denotes the features of the nodes within the neighbourhood $\mathcal{N}(i)$ of node $i$. Detailed, model-specific formulations of Eq. 2 are provided in Sect. B of the appendix.

Eq. 2 implies that the effect of $\epsilon$ can be exacerbated by the potentially large and diverse neighbourhoods $\mathcal{N}(i)$, thereby making the output highly sensitive to changes in $W$. In other words, the effect of weight perturbation can vary dramatically based on the node's connectivity and the characteristics of its neighbours. Such variability leads to significant and less predictable changes in output, illustrating the particular vulnerability of GNNs to parameter mismatches.

The integration of Eq. 1 with Lemma 4.1 and Eq. 2 further gives rise to the following conjecture:

**Conjecture 4.2** (Topology-dependent Complexity in GNNs)**.** *The identification of optimal permutation matrices $\mathbf{P}^*$ for GNNs presents increased complexity compared to the Euclidean domain, contingent upon the topological characteristics inherent to each graph.*

Conjecture 4.2 highlights the essential need for developing GRAMA methods that are specifically tailored to accommodate the unique topologies of graphs, motivating the design of the proposed approach in Sect. 5.

## 5 Proposed Approach: Dual-Message Coordinator and Calibrator

### 5.1 Overview

Motivated by Conjecture 4.2, we introduce in this section the proposed *Dual-Message Coordination and Calibration (DuMCC)* methodology, which is specifically designed to harness the unique topological features of input graphs for achieving GRAMA without relying on human annotations.

The proposed DuMCC is composed of two strategic schemes. In particular, the first *Parent Message Coordination (PMC)* scheme effectively integrates topological information by deriving optimal permutation matrices from layer-specific aggregation results. However, both empirical and theoretical analyses reveal a reduction in node feature variance in child GNNs, suggesting that models derived through this coordination are more susceptible to over-smoothing compared to their parent GNN counterparts.

To address this issue, we further propose the *Child Message Calibration (CMC)* scheme as our second strategic component. This scheme aims to maintain message variance consistency from the parent models, ensuring the retention of feature diversity essential for robust GNN performance. Further elaboration on each component is provided in subsequent sections.

### 5.2 Parent Message Coordination Scheme

Motivated by Conjecture 4.2, which highlights the significance of incorporating topology information in the GRAMA process, we propose a *Parent Message Coordination (PMC)* scheme for identifying

optimal topology-aware permutation matrices $\mathbf{P}^*$ described in Eq. 1. Unlike the vanilla method of VAPI in Sect. 4 that minimises the distance between weight vectors without considering the input graphs' topologies, PMC optimises $\mathbf{P}^*$ by leveraging the topology information embedded in *aggregated messages* from two pre-trained parent GNN models.

Assume we have two pre-trained parent GNNs, denoted by $\mathcal{G}_a$ and $\mathcal{G}_b$, both sharing identical architectures. Their corresponding weight matrices are denoted by $W_a$ and $W_b$, respectively. To establish correspondence between neurons in $W_a$ and $W_b$ as described in Eq. 1, our PMC optimises the permutation matrix $P^{(\ell)}$ at layer $\ell$ by aligning the aggregated messages of the two parent GNNs. To align with the notations [1, 33] used in model merging within the Euclidean domain, the associated optimisation process can be formulated as follows:

$$
\begin{aligned}
P^{(\ell)} = \arg \min_{P^{(\ell)} \in \mathbf{P}^*} \sum_i \big\| \operatorname*{Agg}_{j \in \mathcal{N}(i)} \phi\left(W_a^{(\ell-1)}; X_{j,a}^{(\ell-1)}, e_{ij}\right) \\
- P^{(\ell)} \cdot \operatorname*{Agg}_{j \in \mathcal{N}(i)} \phi\left(W_b^{(\ell-1)}; X_{j,b}^{(\ell-1)}, e_{ij}\right) \big\|^2,
\end{aligned}
\tag{3}
$$

where $\phi$ denotes the message function that encodes both node features $X$ and edge features $e$, and Agg represents the message aggregation function that accumulates incoming messages from $\phi$, acting on each node $i$. $\mathcal{N}(i)$ denotes the set of neighbours of node $i$. The minimisation problem in the form of Eq. 3 can be typically transformed into a maximisation problem to maximise an inner product (as derived from expanding Eq. 3), thereby fitting it within the framework of a standard linear assignment problem, as also done in the works of [1, 33, 35, 44].

Eq. 3 is based on the rationale that aggregated messages inherently encapsulate essential graph topologies, and that structurally similar GNNs typically generate analogous aggregated messages when tasked with similar graph operations and topologies. As such, through Eq. 3, $\mathbf{P}^*$ can be determined in a topology-aware manner, matching aggregated messages to reflect the topological characteristics of the graphs. Here, we clarify that while our approach involves passing the graph data to the pre-trained model to capture the graph-specific topological characteristics by utilising $X$, it requires only a single forward pass of the unlabelled graph data to extract messages for alignment—eliminating the need for iterative training or ground-truth labels. Subsequently, child GNNs can be derived through linear parameter interpolation.

Despite the encouraging performance, we observe that the child GNN, derived from the proposed PMC, exhibits a reduction in the variance of node embeddings. We conjecture that this reduction stems from an averaging effect, which may smooth out the distinctive features captured by each parent model, particularly when these models have learned different structural aspects of the graph:

**Lemma 5.1** (Variance Reduction in Interpolated Graph Embeddings). *The variance of the graph embeddings in an interpolated child GNN is typically smaller than the variances of the embeddings from the individual pre-trained parent GNNs.*

The full proof of Lemma 5.1 is detailed in Appendix G.2, providing evidence of feature homogenisation within child GNNs. In the context of GNNs, we further explore and establish the following proposition:

**Proposition 5.1** (Increased Susceptibility to Over-Smoothing in Child GNNs). *Interpolated child GNNs exhibit increased susceptibility to over-smoothing compared to their parent networks, as measured by Dirichlet energy.*

The detailed proof of Proposition 5.1 is provided in Appendix G.3, utilising the quantitative over-smoothing measurement based on Dirichlet energy [41], where a lower value indicates greater homogeneity or smoothness among node features. In particular, our theoretical analysis in Appendix G.3 demonstrates that:

$$
\mathcal{E}(X^\ell) \leq \max\left(\mathcal{E}(X_a^\ell), \mathcal{E}(X_b^\ell)\right),
\tag{4}
$$

where $\mathcal{E}(X^\ell)$ denotes the Dirichlet energy for the node features $X^\ell$ at layer $\ell$ of the child GNN.

In GRAMA, the parameter $\alpha$ is typically set to 0.5 to ensure unbiased knowledge integration from both pre-trained models. This setting promotes a balanced contribution from each model and prevents any bias toward the characteristics of one over the other. As detailed in Appendix G.3, with $\alpha$ at this level, the Dirichlet energy of the interpolated child GNN, $\mathcal{E}(X^\ell)$, significantly decreases

compared to the Dirichlet energies of the individual parent models, indicating a higher susceptibility to over-smoothing.

Corresponding to this theoretical analysis, empirical evidence is presented in Fig. 1, where 60 parent models are pre-trained on distinct partitions of the `ogbn-products` dataset with different random seeds. Additional implementation details are provided in the appendix. Fig. 1 further demonstrates the increased smoothing effect, which can potentially diminish the model's expressive power and discriminative capability.

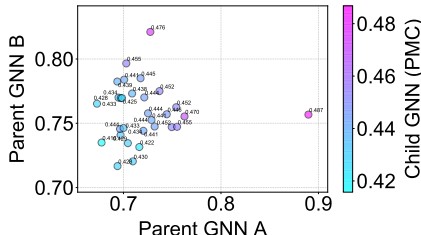

Figure 1: Comparison of *Dirichlet energies* between pre-trained parent GNNs and the corresponding child from PMC.

### 5.3 Child Message Calibration Scheme

To mitigate the over-smoothing issue identified in Proposition 5.1, one potential solution involves leveraging established methods designed to address over-smoothing, such as PairNorm [65] and residual connections [32]. However, to the best of our knowledge, all these existing solutions typically require re-training the model. This requirement contradicts the fundamental principle of our GRAMA approach, which aims for training-free model reuse.

To address this issue, we introduce a *Child Message Calibration (CMC)* scheme designed to refine the message statistics of the obtained child GNN without the need for re-training or ground-truth labels. Central to this scheme is our *Learning-Free Message Normalisation (LFNorm)* layer specifically tailored for our GRAMA task, inspired by [28]. This layer is intended to enhance the discriminative power and representational capacity of the child GNN by promoting a more diverse node feature distribution.

According to Proposition 5.1, while the shift in mean node features is typically less problematic than variance reduction, it is noted in [6] that mean statistics in GNNs also carry vital graph structural information. Therefore, our approach aims to simultaneously refine both mean and variance statistics in the child GNN, ensuring a comprehensive enhancement of topological representation.

Towards this end, we first process the raw graph through the parent GNNs to compute the target message mean and variance intended for alignment in the child GNN. Subsequently, we integrate the LFNorm layer into the child GNN to refine the message statistics using the derived mean and variance from the parent models:

$$\tilde{m}_{ij} = \sqrt{\frac{\text{Var}(m_{ij}^a) + \text{Var}(m_{ij}^b)}{2}} \cdot \left( \frac{m_{ij} - \mu_{\mathcal{N}(i)}}{\sigma_{\mathcal{N}(i)}} \right) + \frac{\text{E}(m_{ij}^a) + \text{E}(m_{ij}^b)}{2}, \tag{5}$$

where $m_{ij} = \phi_{j \in \mathcal{N}(i)}(W^{(\ell-1)}; X_j^{(\ell-1)}, e_{ij})$ represents the messages from node $j$ to node $i$, as specified in Eq. 3. In this setup, $\tilde{m}_{ij}$ denotes the statistically calibrated message within the child GNN, while $\mu_{\mathcal{N}(i)}$ and $\sigma_{\mathcal{N}(i)}$ represent the mean and standard deviation, respectively, of the messages directed to node $i$ in the child GNN.

Eq. 5 ensures that the normalised messages in the interpolated child model maintain a balanced representation of central tendencies from both pre-trained parent models. This method effectively reduces the risk of over-smoothing in child GNNs by preserving essential topological statistics from the parent models.

A more detailed algorithmic procedure is outlined in Alg. 1. In practice, we find that incorporating a single LFNorm layer and aligning the overall message mean and variance of the parent GNNs is typically sufficient to achieve favourable performance while minimising computational costs.

## 6 Experiments

We evaluate the performance of DuMCC across seven benchmarks spanning five GNN architectures. More ablation studies and sensitivity analyses, additional results and implementation details, as well as more visualisations, are detailed in Secs. D and E of the appendix.

**Algorithm 1** The proposed Dual-Message Coordinator and Calibrator (DuMCC) for GRAMA.

---

**Input:** Pre-trained Parent GNNs $\mathcal{G}_a$ and $\mathcal{G}_b$, interpolation factor $\alpha$.
**Output:** Child GNN $\mathcal{G}$ that integrates the expertise of $\mathcal{G}_a$ and $\mathcal{G}_b$ in a learning-free manner.

```
// Parent Message Coordination
```
**foreach** *layer $\ell$ from 1 to L* **do**

 Extract weights $W_a^{(\ell)}$ from $\mathcal{G}_a$; Extract weights $W_b^{(\ell)}$ from $\mathcal{G}_b$

```
   // Compute permutation matrix for current layer with aggregated messages
```
$$P^{(\ell)} \leftarrow \arg\min_{P^{(\ell)} \in \mathbf{P}^*} \sum_i \| \underset{j \in \mathcal{N}(i)}{\text{Agg}} \; \phi\left(W_a^{(\ell-1)}; X_{j,a}^{(\ell-1)}, e_{ij}\right)$$
$$- P^{(\ell)} \cdot \underset{j \in \mathcal{N}(i)}{\text{Agg}} \; \phi\left(W_b^{(\ell-1)}; X_{j,b}^{(\ell-1)}, e_{ij}\right) \|^2$$

```
   // Interpolate weights for current layer
```
$$W^{(\ell)} \leftarrow \alpha W_a^{(\ell)} + (1-\alpha) P^{(\ell)} W_b^{(\ell)} (P^{(\ell-1)})^T$$

**end**

$\mathcal{G} \leftarrow \{W^{(\ell)}\}_{\ell=1}^L$

```
// Child Message Calibration
```
**foreach** *layer $\ell$ from $(L-n)$ to L* **do**
 **foreach** *edge $(i,j)$ in the graph* **do**
```
      // Compute the messages in Ga and Gb
```
  $m_{ij}^a \leftarrow \phi_{j \in \mathcal{N}(i)}(W_a^{(\ell-1)}; X_{j,a}^{(\ell-1)}, e_{ij}); \; m_{ij}^b \leftarrow \phi_{j \in \mathcal{N}(i)}(W_b^{(\ell-1)}; X_{j,b}^{(\ell-1)}, e_{ij})$
```
      // Compute the message for G
```
  $m_{ij} \leftarrow \phi_{j \in \mathcal{N}(i)}(W^{(\ell-1)}; X_j^{(\ell-1)}, e_{ij})$
```
      // Compute the scale and shift parameters
```
  $\beta \leftarrow \sqrt{\left(\text{Var}(m_{ij}^a) + \text{Var}(m_{ij}^b)\right)/2}; \; \gamma \leftarrow \left(\text{E}(m_{ij}^a) + \text{E}(m_{ij}^b)\right)/2$
```
      // Learning-free message calibration for G
```
  $\tilde{m}_{ij} \leftarrow \beta\left((m_{ij} - \mu_{\mathcal{N}(i)})/\sigma_{\mathcal{N}(i)}\right) + \gamma$
 **end**
 **foreach** *node $i$ in the graph* **do**
```
      // Aggregate the calibrated messages and update features
```
  $X_i^\ell \leftarrow \text{Agg}_{j \in \mathcal{N}(i)} \tilde{m}_{ij}$
 **end**
**end**

---

**Implementation Details.** Detailed dataset descriptions and statistics are provided in Appendix C. For multi-class classification tasks on ogbn-arxiv [17], ogbn-products [4], and ModelNet40 [55], we adopt the dataset partition strategy widely used in model merging within the Euclidean domain [1, 28]. Specifically, each dataset is randomly split into two disjoint subsets: the first subset comprises 20% of the data with odd labels and 80% with even labels, while the second subset is arranged vice versa.

Table 2: Multi-class molecule property prediction results for parent GNNs, each pre-trained on disjoint partitions of the `ogbn-arxiv` and `ogbn-products` datasets [18].

| Methods | Re-train? | ogbn-arxiv | | ogbn-products | |
|---|---|---|---|---|---|
| | | Dataset A | Dataset B | Dataset C | Dataset D |
| Parent GCN A [30] | - | 0.7193 | 0.5516 | N/A | N/A |
| Parent GCN B [30] | - | 0.6564 | 0.7464 | N/A | N/A |
| Parent GraphSAGE C [13] | - | N/A | N/A | 0.7982 | 0.7308 |
| Parent GraphSAGE D [13] | - | N/A | N/A | 0.7626 | 0.7904 |
| KA [25] (Section 3) | √ | 0.7150 | 0.6687 | 0.7973 | 0.7775 |
| VPI [54] (Section 4) | × | 0.3486 | 0.4361 | 0.6568 | 0.6546 |
| VAPI [1] (Section 4) | × | 0.6140 | 0.5752 | 0.5425 | 0.5779 |
| Ours (w/o CMC) | × | 0.6531 | 0.5957 | 0.7374 | 0.7414 |
| **Ours (w/ CMC)** | × | **0.6645** | **0.6382** | **0.7647** | **0.7515** |

For the semantic segmentation task on S3DIS [3], we train the two parent models using Areas 1, 2, 3 and Areas 2, 3, 4, 6, respectively, with Area 5 designated for testing, as also done in [32]. In the multi-label classification task on ogbn-proteins [17], one parent model is trained on nodes with odd labels and the other on nodes with even labels. Implementation follows the official codes provided by the Deep Graph Library (DGL) [52] and the original authors, including detailed architectures and hyperparameter settings. We set the interpolation factor $\alpha$ in Eq. 1 to 0.5 for all experiments, with a sensitivity analysis provided in Sect. D of the appendix. For models originally equipped with

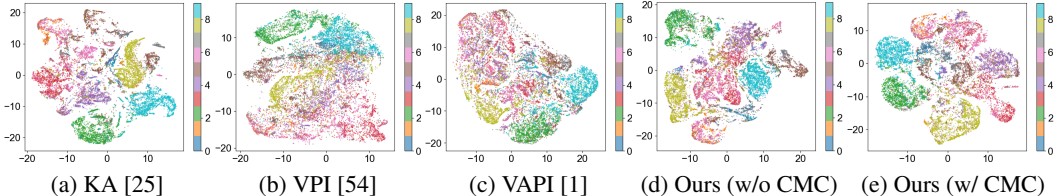

| (a) KA [25] | (b) VPI [54] | (c) VAPI [1] | (d) Ours (w/o CMC) | (e) Ours (w/ CMC) |

Figure 2: The t-SNE visualisations of various methods on a subset comprising the first 10 classes of `ogbn-arxiv`. Additional visualisations for the remaining classes are available in Appendix E.

| Methods | Re-train? | Dataset I | Dataset J |
|---|---|---|---|
| Parent DGCNN I [53] | - | 0.9159 | 0.8151 |
| Parent DGCNN J [53] | - | 0.8862 | 0.9275 |
| KA [25] (Section 3) | √ | 0.9250 | 0.9283 |
| VPI [54] (Section 4) | × | 0.4518 | 0.4096 |
| VAPI [1] (Section 4) | × | 0.6538 | 0.5482 |
| Ours (w/o CMC) | × | 0.8326 | 0.8088 |
| **Ours (w/ CMC)** | × | **0.8920** | **0.8574** |

Table 4: Results of the point cloud classification task on ModelNet40 [55] using DGCNN, with two parent models trained on disjoint partitions.

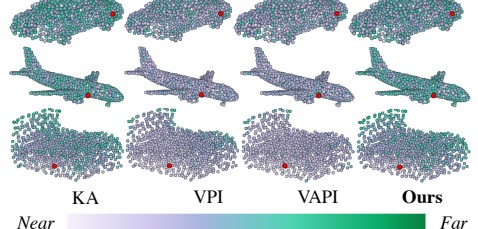

KA    VPI    VAPI    **Ours**

*Near*                        *Far*

Figure 3: Visualisations of feature space structures, depicted by the distances between the red point and all other points.

normalisation layers, we recompute the running mean and running variance for the student GNN. In particular, the recomputation of statistics is performed concurrently with that of the LFNorm layer for the child GNN in CMC. More implementation details are elaborated in Sect. E of the appendix.

**Comparison Methods.** Considering the limited exploration of multi-GNN reuse in existing literature, we focus our comparison of the proposed DuMCC approach on the training-dependent KA [25] and training-free VPI [54] and VAPI [1] methods, as introduced in Sect. 4. Furthermore, in Sect. E of the appendix, we also provide the results of retraining a child model using combined parent datasets with ground-truth labels, which establish an upper bound for GRAMA's performance.

**Node Property Prediction.** Tab. 2 presents the results for the multi-class node classification task. The proposed DuMCC framework, as shown in the table, achieves a more balanced performance across all datasets. It notably outperforms the parent models on datasets that were not used for their training. Additionally, the last two lines of Tab. 2 detail an ablation study on the proposed CMC scheme, demonstrating its ability to enhance performance beyond PMC. While our method slightly lags behind KA in performance, KA involves a complex re-training process, whereas our approach is completely training-free. A sample t-SNE visualisation of the results is provided in Fig. 2.

We further show in Tab. 3 the results of multi-label molecule property prediction. Notably, our approach slightly outperforms KA in Tab. 3, underscoring the limitations of KA discussed in Sect. 3. These limitations stem from KA's reliance on soft labels produced by the teacher GNNs, making it vulnerable to their misclassification errors. Furthermore, we explore the potential of multi-model GRAMA by concurrently reusing three pre-trained node classification GNNs, as discussed in Sect. F of the appendix.

**Graph Property Prediction.** Tab. 3 also presents results for the graph classification task using GAT architectures. The proposed DuMCC demonstrates enhanced equilibrium in performance relative to the parent models. Notably, DuMCC slightly outperforms KA on ogbg-molbace, further illustrating

Table 3: Results for multi-label node classification and graph classification, indicating KA's vulnerability to misclassification errors from pre-trained models.

| Architectures | GIN [56] | | | | Architectures | GAT [48] | | | |
|---|---|---|---|---|---|---|---|---|---|
| Methods | Parent E | Parent F | KA | Ours | Methods | Parent G | Parent H | KA | Ours |
| `ogbn-proteins` | 0.7478 | 0.7222 | 0.7215 | 0.7341 | `ogbg-molbace` | 0.7247 | 0.4067 | 0.6135 | 0.6296 |
| | | | | | `ogbg-molbbbp` | 0.4681 | 0.6366 | 0.6446 | 0.5275 |

Table 5: Results of the 3D semantic segmentation task on the S3DIS dataset [3], with detailed per-class results provided. Architecture details can be found in Sect. E of the appendix.

| Methods | Re-train? | Structural Elements | | | | | | | |
|---|---|---|---|---|---|---|---|---|---|
| | | ceiling | floor | wall | beam | column | window | door | mean |
| Parent DGCNN K [53] | - | 0.9655 | 0.9947 | 0.9355 | 0.0079 | 0.0557 | 0.4529 | 0.1430 | 0.5079 |
| Parent DGCNN L [53] | - | 0.9529 | 0.9927 | 0.9546 | 0.0573 | 0.0661 | 0.3555 | 0.1335 | 0.5018 |
| KA [25] (Section 3) | √ | 0.9580 | 0.9943 | 0.9003 | 0.0681 | 0.1835 | 0.5154 | 0.7048 | 0.6178 |
| VPI [54] (Section 4) | × | 0.6909 | 0.9871 | 0.3612 | 0.0000 | 0.0044 | 0.0000 | 0.0037 | 0.2925 |
| VAPI [1] (Section 4) | × | 0.5338 | 0.8766 | 0.6825 | 0.0382 | **0.0046** | 0.0284 | 0.0019 | 0.3094 |
| Ours (w/o CMC) | × | **0.9804** | **0.9967** | 0.9186 | 0.0302 | 0.0000 | 0.0008 | 0.0752 | 0.4289 |
| **Ours (w/ CMC)** | × | 0.9695 | 0.9962 | **0.9290** | **0.1466** | 0.0004 | **0.0344** | **0.1047** | **0.4544** |

| Methods | Re-train? | Furniture | | | | | | Others | Overall |
|---|---|---|---|---|---|---|---|---|---|
| | | table | chair | sofa | bookcase | board | mean | clutter | mean |
| Parent DGCNN K [53] | - | 0.7182 | 0.7746 | 0.0216 | 0.4980 | 0.4022 | 0.4829 | 0.6515 | 0.8181 |
| Parent DGCNN L [53] | - | 0.7447 | 0.9255 | 0.1301 | 0.5267 | 0.1774 | 0.5009 | 0.6038 | 0.8174 |
| KA [25] (Section 3) | √ | 0.7358 | 0.8649 | 0.0295 | 0.5377 | 0.4211 | 0.5178 | 0.6844 | 0.8382 |
| VPI [54] (Section 4) | × | 0.0159 | 0.2802 | 0.0019 | 0.0135 | 0.0001 | 0.0623 | 0.4188 | 0.4791 |
| VAPI [1] (Section 4) | × | 0.0117 | 0.1984 | **0.0142** | 0.1235 | 0.0025 | 0.0701 | 0.2326 | 0.5060 |
| Ours (w/o CMC) | × | 0.5953 | **0.8485** | 0.0015 | 0.1160 | 0.0009 | 0.3125 | **0.5279** | 0.7497 |
| **Ours (w/ CMC)** | × | **0.6351** | 0.8089 | 0.0086 | **0.2867** | **0.0182** | **0.3515** | 0.5274 | **0.7676** |

that KA is vulnerable to the errors of pre-trained models. In contrast, our approach does not rely on soft labels, thus avoiding this limitation.

**3D Object Recognition and Semantic Parsing.** Tab. 4 and Fig. 3 illustrate the quantitative results and qualitative visualisations for the point cloud classification task, respectively. The proposed DuMCC outperforms model I on dataset J and model J on Dataset I without requiring re-training. Moreover, Tab. 4 demonstrates that our approach significantly outperforms two vanilla methods. Fig. 3 further illustrates the structure of the feature space, revealing that our method produces semantically similar structures to those achieved by KA with re-training. We also show in Tab. 5 the results for the large-scale indoor semantic segmentation task. Our method notably surpasses other learning-free GNN reuse methods VPI and VAPI. Further qualitative and quantitative results across various dataset splits and network architectures are provided in Sect. E of the appendix.

## 7 Conclusions and Limitations

In this paper, we explore a novel GRAMA task for learning-free GNN reuse. The child model from GRAMA is expected to functionally merge knowledge from pre-trained parent models. Uniquely, GRAMA establishes the first paradigm in GNN reuse that operates entirely without re-training or fine-tuning, while also eliminating the need for ground-truth labels. To this end, we start by developing two vanilla GRAMA approaches, which reveal specific challenges inherent to GRAMA. These challenges motivate us to develop a DuMCC framework for topology-aware model reuse, leveraging a parent message coordination scheme followed by child message calibration. Experiments on node- and graph-level tasks across various domains demonstrate the effectiveness of the proposed approach for annotation-free knowledge transfer without additional learning.

Despite its strengths, the proposed DuMCC is primarily designed for homogeneous GRAMA, as discussed in Sect. 3. Currently, the framework does not support cross-architecture heterogeneous GRAMA, where parent models have different architectures, such as a combination of GCN and GraphSAGE. Additionally, it does not handle scenarios where parent models address tasks at different levels, such as node-level versus graph-level tasks—another aspect of heterogeneous GRAMA. These limitations primarily arise from the absence of direct correspondence between the differing architectural layers of the parent models, an issue we plan to explore in our future work. We will also explore the possibility of a fully data-independent GRAMA scheme and investigate broader applications beyond training-free model reuse, such as its use as a pre-processing step to facilitate graph-based knowledge amalgamation. Further discussions on limitations and potential solutions are provided in Sect. H of the appendix.

## Acknowledgement

This research / project is supported by the National Research Foundation, Singapore, and Cyber Security Agency of Singapore under its National Cybersecurity R&D Programme and CyberSG R&D Cyber Research Programme Office, as well as Australian Research Council Discovery Project DP190103301.

Any opinions, findings and conclusions or recommendations expressed in these materials are those of the author(s) and do not reflect the views of National Research Foundation, Singapore, Cyber Security Agency of Singapore as well as CyberSG R&D Programme Office, Singapore.

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
