# OpenReview forum: "Deep Graph Mating"
_NeurIPS.cc/2024/Conference — NeurIPS 2024 poster_

### Official Review · Reviewer_tRzs · 2024-07-05

**Soundness:** 3
**Presentation:** 3
**Contribution:** 3
**Rating:** 6
**Confidence:** 5

**Summary:**

In this paper, the authors present a method for learning-free and label-free model reuse for graph neural networks. They dub the task Deep Graph Mating (Grama). Without relying on costly fine-tuning or re-training, Grama aims to generate a child model by reusing and fusing knowledge from pre-trained parent models, particularly by managing pre-trained parameters. Compared with conventional model reuse tasks (knowledge distillation, and knowledge amalgamation) in the GNN domains, the authors claim in Tab 1 that Grama simultaneously enables multi-model reuse, annotation free, and training/fine-tuning free, leading to more resource-efficient model reuse. To achieve Grama, the authors begin by deriving two vanilla Grama methods, dubbed VPI and VAPI, motivated by the permutation invariance proposition in GNNs. However, both VPI and VAPI don’t work well, and the authors observe unique challenges for Grama, namely the increased sensitivity to weight misalignment and the accompanying topology-dependent complexities, substantiated by a series of propositions, lemmas, and conjectures. Motivated by the identified challenges, the authors introduce the proposed DuMCC framework, which includes a PMC and CMC. PMC is designed to identify topology-aware permutation matrices but suffered from increased susceptibility to over-smoothing. To solve this problem, the authors further designs CMC, which is to refine message statistics of the child network through a learning-free message normalization layer. Extensive experiments (7 benchmarks, 4 tasks, 5 architectures) in various graph applications have been performed to validate the effectiveness of the proposed framework, covering popular GNNs including GCN, GraphSAGE, GAT, GIN, and DGCNN. The authors also provide extensive supplementary information with detailed proofs in the appendix.

**Strengths:**

The positive traits of this paper include:

- Well-organized paper with a very clear logic flow. The authors first give a well-defined problem of Grama with reasonable motivations. Then they present two vanilla methods as initial solutions to solve the Grama problem. Based on these methods, the authors identify the unique challenges from Grama with elaborated analysis and clear thoughts. Motivated by the identified challenges, a DuMCC framework is proposed. The way the authors present the entire paper make it very easy for following its development.

- Interesting idea. To the best of my knowledge, Grama is the first to simultaneously enable multi-model reuse, annotation-free, and training/fine-tuning free capabilities. The authors clearly demonstrate their idea by comparing Grama with previous model reuse methods in Tab 1, which is very useful for understanding the main benefits of Grama. It is also interesting to explore the correspondence in the weight space for different GNNs, which also appears to be the first exploration in this field of GNN.

- Solid intuitions and well-established algorithm. At first sight, the proposed approach appears a bit ad-hoc. However, the authors provide sufficient motivations for each component in DuMCC, making their method well-justified. This includes extensive discussions on vanilla methods and the over-smoothing issues from PMC, with a motivating example shown early in Fig 1, making the proposed method technically sound.

- Experiments are conducted thoroughly. It is clear that the authors try to cover a wide range of tasks and various GNN architectures in their experiments. The tasks include node property prediction, graph property prediction, 3D object recognition, and 3D semantic segmentation, and the architectures include GCN, GraphSAGE, GAT, GIN, and DGCNN. The effectiveness of the proposed method and nearly all claims in the paper are thoroughly verified.

- Good reproducibility. Extensive supplementary details are provided in the appendix, with source code and models provided in the supplementary material. Algorithm 1 is clear and useful for capturing the main procedure of the entire framework.

**Weaknesses:**

- In Sec 5.3, the authors introduce a learning-free message normalization (LFNorm) layer for statistics calibration in CMC, which is technically sound. However, the authors miss the discussions on the Grama case where pre-trained models already contained normalization layers, which is common in the area of graph neural networks. In that case, it is unclear whether the batch statistics are also recomputed in CMC. If they are, it should be specified when the recomputation occurs: simultaneously with LFNorm or after?

- Because the authors have claimed multimodel reuse as a contribution of this work, I would advise the authors to include more experiments for the Grama cases with three or more pre-trained models. Otherwise, clear statements should be added to limit the scope of the study to the reuse of two pre-trained models.

- The descriptions and explanations in the experimental results are not closely matched with the authors’ assumptions and propositions from the previous sections. It is suggested that the authors more closely relate the texts in the experimental section to the methods section, particularly by specifying which results validate which proposition or claim. This would make the paper more readable.

- In the appendix, there is a minor problem where Eq 30 extends beyond the margin of the paper.

**Questions:**

1. Can you clarify if the statistics in the original normalization layer are recomputed during Grama? If so, when would they be recomputed?

2. Have you experimented with using three or more pre-trained models in Grama?

3. Consider improving the experimental section to more closely relate the texts to the validated claims.

Although the authors missed some key information, I feel these could be easily addressed. On balance, the paper is both novel and useful. The concept of training-free and label-free model reuse for GNN had not been done before.

**Limitations:**

The paper has thoroughly discussed limitation in Sec 7 and social impact in appendix.

---

> ### Author Rebuttal · Authors · 2024-08-06
>
> ### **Response to Reviewer tRzs**
>
> We appreciate the reviewer for the positive support and constructive comments.
>
> `W1. & Q1.` **Statistics recomputation details**
>
> >"The authors miss the discussions on the Grama case where pre-trained models already contained normalization layers, which is common in the area of graph neural networks. In that case, it is unclear whether the batch statistics are also recomputed in CMC. If they are, it should be specified when the recomputation occurs: simultaneously with LFNorm or after?"
>
> >"Can you clarify if the statistics in the original normalization layer are recomputed during Grama? If so, when would they be recomputed?"
>
> `Response:` We apologise for not having sufficiently highlighted the details regarding the recomputation of statistics, which might have led to them being overlooked by the reviewer. In fact, as detailed in line 313-314, we have indeed discussed the scenario where models are equipped with normalisation layers. We have explicitly explained that the running mean and variance of the original normalisation layers are recalculated for the child GNN. In our revised version, we will enhance the clarity of this detail further by also providing additional explanations in Sect. 5.3.
>
> But admittedly, we apologise for not having specified when the recomputation of statistics occurs for the original normalisation layers in our paper. We appreciate the reviewer's insightful comments on this issue. As specified in the source code provided in the supplementary material, we clarify that the recomputation of statistics is performed concurrently with that of the LFNorm layer for the child GNN. In the revision, we will provide a detailed clarification and add the discussion on the normalisation-related procedures described above in Sect. 5.3 and Sect. 6.
>
> ---
>
> `W2. & Q2.` **Multi-model GRAMA results**
>
> >"Because the authors have claimed multimodel reuse as a contribution of this work, I would advise the authors to include more experiments for the Grama cases with three or more pre-trained models. Otherwise, clear statements should be added to limit the scope of the study to the reuse of two pre-trained models."
>
> >"Have you experimented with using three or more pre-trained models in Grama?"
>
> `Response:` We appreciate the reviewer's constructive suggestion. As suggested, we have conducted additional experiments on disjoint partitions of the ogbn-arxiv dataset using the architecture described in Tab. 10 of the main paper, specifically for the proposed GRAMA in scenarios involving the simultaneous reuse of three pre-trained models. The results, presented below, demonstrate that our proposed DuMCC approach is also effective in multi-model GRAMA contexts. These results will be incorporated into the revised version of the paper.
>
> | Models | Performance: Dataset 1 | Performance: Dataset 2 |  Performance: Dataset 3 |
> |:----- | :----- | :----- | :----- |
> | Parent 1 | 0.6645 | 0.6476 | 0.6040 |
> | Parent 2 | 0.4796  | 0.7044 | 0.4651 |
> | Parent 3 | 0.5805 | 0.6078 | 0.7728 |
> | Child    | 0.5574 | 0.6360 | 0.5478 |
>
>
> ---
>
> `W3. & Q3.` **Linking experiments to the claims to validate**
>
> >"The descriptions and explanations in the experimental results are not closely matched with the authors' assumptions and propositions from the previous sections. It is suggested that the authors more closely relate the texts in the experimental section to the methods section, particularly by specifying which results validate which proposition or claim. This would make the paper more readable."
>
> >"Consider improving the experimental section to more closely relate the texts to the validated claims."
>
> `Response:` We would like to thank the reviewer for the constructive suggestion. In our revised version, we will strive to expand the text in Sect. 6. This will include more detailed descriptions that closely and explicitly connect the results presented in Figs. 2-3 and Tabs. 2-5 with the validated conjectures, propositions, and claims discussed in Sects. 4-5.
>
> ---
>
> `W4.` **Minor problem**
>
> >"In the appendix, there is a minor problem where Eq 30 extends beyond the margin of the paper."
>
> `Response:` Thank you for pointing out this formatting issue. We will address this issue in the revision by breaking Eq. 30 into three separate lines to ensure it fits within the margins.

---

### Official Review · Reviewer_Uv5e · 2024-07-08

**Soundness:** 3
**Presentation:** 4
**Contribution:** 3
**Rating:** 7
**Confidence:** 5

**Summary:**

This paper proposes Deep Graph Mating, a novel task for model reuse in non-Euclidean domains, specifically focusing on GNNs. The goal is to create a child GNN that combines knowledge from pre-trained parent GNNs without requiring re-training, fine-tuning, or ground-truth labels. The process firstly identifies the permutation invariance properties of GNNs, and accordingly explores two naïve methods: Vanilla Parameter Interpolation (VPI) and Vanilla Alignment Prior to Interpolation (VAPI), both of which were shown to be insufficient due to challenges such as parameter misalignment and topological dependencies.

The contribution of this paper is to address this issue by proposing a Dual-Message Coordination and Calibration (DuMCC) methodology that optimizes the permutation matrices for parameter interpolation (PMC scheme) by incorporating topological information, and solves over-smoothing by calibrating the message statistics for child GNNs (CMC scheme). This results in performance on par with traditional re-training methods but without the associated training cost, which is validated on seven datasets across node and graph classification as well as semantic parsing tasks. Code and proofs are given in the appendix and the supplementary material.

**Strengths:**

I see several merits of this paper:

Addressing the challenge of model reuse in GNNs is a significant topic, especially given the increasing scale of graph data and models in this era. The proposed GRAMA bypasses the need for re-training or fine-tuning, which is commonly required in existing model reuse approaches. GRAMA also removes the need for ground-truth labels, enhancing its generalizability in scenarios where such labels are unavailable. These properties make GRAMA suitable for graph analysis scenarios where resources are limited.

The authors present a couple of effective schemes PMC and CMC to deal with the challenges in GRAMA, specifically focusing on parent models and child model. These proposed methods seem overall straightforward yet supported by strong motivations and solid theoretical analysis, such as the amplified sensitivity to parameter misalignment and the increased susceptibility to over-smoothing in child models. The paper also conducts a thorough analysis of these methods across seven graph benchmarks and demonstrates good performance. I see competitive values in Tables 4,5 for the large-scale point cloud benchmarks, without needing to re-training.

Overall, the paper is well-written and the main ideas, arguments, and algorithm are very easy to follow, with ample details provided in the appendix.

**Weaknesses:**

I see the following weaknesses in this paper:

While I appreciate the authors’ attempt to present the complex trajectory of the paper (from defining the problem and elaborating the task motivation, to developing two naïve methods and analyzing challenges, and finally proposing two schemes) in a narrative style for easier understanding, there is an imbalance in the organization of content. The current organization of the paper only allocates less than two pages to the experiment section, leaving many experimental details to the appendix. I suggest the authors move more experimental details from the appendix into the main body of the paper, reducing the current extensive focus on the motivations of the method, and instead, expanding more information about the datasets used, enhancing the details and settings of the comparative methods, and elaborating on the experimental procedures. Also, the connection between the main paper and the appendix is very limited. The authors vaguely direct readers to Section D for further details, without specifying specific tables or paragraphs, which makes it hard to locate relevant information.

Also, the explanation regarding the heterogeneous GRAMA in Line 123 actually makes me confused. The statement is too vague. What exactly are the key differences between the homogeneous and heterogeneous cases of GRAMA? Does it specifically relate to the same or different architectures of the pre-trained models, or does it also involve models specializing in different tasks? For example, if there are two pre-trained models with the same architecture but trained for different tasks, would this scenario be classified as homogeneous or heterogeneous GRAMA? Furthermore, while the authors claim that “our initial investigation in this paper is confined to scenarios where pre-trained GNNs possess identical architectures yet are trained on separate datasets”, the authors should at least provide some insights or potential solutions for solving heterogeneous GRAMA, or perhaps consider removing this statement of heterogeneous GRAMA since this is not the contribution of this paper.

Following this discussion of heterogeneous GRAMA, I think it is at least worth a try to apply GRAMA to two different GNN variants, such as GCN and GraphSage, since the key learnable parameters in these GNNs are all MLPs. The existing literature has demonstrated the importance of MLPs to GNN performance [1]. Exploring this avenue could be very interesting; if successful, the contributions of this paper would be significantly enhanced. Furthermore, the results may shed light on the connections between different GNN variants in the weight space.

In Tables 2 to 5, the authors overlook an important comparative result: the performance of retraining a multi-dataset model that can jointly combine the expertise of both parent models. While this approach involves retraining, it can at least serve as an upper bound for GRAMA, showing the potential space for further improvement.

It is also not very clear to me why the authors chose a 20%/80% splitting ratio in the experiments, as there are too few explanations about the splitting protocol. For example, is the split random? More details should be provided, as this concerns the pre-trained models and can reveal how knowledge from the two pre-trained models is distributed and when the proposed method is effective. Would other splitting, such as 10/90, work for GRAMA? It is necessary to conduct more experiments with more splitting ratios to make the authors’ claims more convincing.

The current discussion of limitations in this paper appears somewhat constrained. The authors should consider expanding this section, either in the main body of the paper or in the appendix. Another limitation of this work, compared to previous approaches such as KA, is the combination of models working on different levels of tasks, such as node classification and graph classification. Weight-space combination for models addressing various tasks has recently been explored in the Euclidean domain such as ZipIt [2], making it intriguing to see whether similar results can also be achieved in the non-Euclidean domain.

Minor issues in the paper:

Line 26: "reduing" $\rightarrow$ "reducing";

Line 106: "presents" $\rightarrow$ "present";

Line 193: "are" $\rightarrow$ "is".

[1] Han, Xiaotian, et al. MLPInit: Embarrassingly Simple GNN Training Acceleration with MLP Initialization. ICLR 2023.

[2] Stoica, George, et al. ZipIt! Merging Models from Different Tasks without Training. ICLR 2024.

**Questions:**

See weaknesses. Overall, I believe this paper addresses a significant problem and proposes a novel approach for model reuse in GNNs. The motivation is strong, supported by solid theoretical analysis. Despite the merits, I see several weaknesses. Therefore, I rate the paper as borderline at this stage and look forward to the authors' responses and further discussions regarding my concerns in the weaknesses section.

**Limitations:**

Yes, limitations and broader impacts have been discussed. An additional limitation to consider would be the scenario where two pre-trained models address tasks at different levels, such as node and graph-level tasks.

---

> ### Author Rebuttal · Authors · 2024-08-06
>
> ### **Response to Reviewer Uv5e (Part 1/2)**
>
> We truly appreciate the reviewer's insightful comments, and would like to address them as follows. Due to character limitations, we have to split our response into two parts. The second part will be provided as a comment following our initial response.
>
> `W1.` **Imbalanced organisation**
>
> >"The current organization of the paper only allocates less than two pages to the experiment section, leaving many experimental details to the appendix."
>
> `Response:` We appreciate the reviewer for the suggestion. Given the novelty of the studied GRAMA task and the sophisticated rationale behind the proposed DuMCC, our organisational approach was intended to provide detailed explanations, thus facilitating easier comprehension for the readers. These include:
> - `Sect. 3`: Motivation, definition, and applications of the GRAMA task;
> - `Sect. 4.1`: Vanilla solutions for the GRAMA task described in `Sect. 3`;
> - `Sect. 4.2`: Challenges associated with the vanilla solutions from `Sect. 4.1`, leading to the motivations for our proposed DuMCC in `Sect. 5`;
> - `Sect. 5.1`: An overview of our proposed DuMCC, motivated by the challenges outlined in `Sect. 4.2`;
> - `Sect. 5.2`: The first component of DuMCC, PMC, including an analysis of its shortcomings, which subsequently motivates the introduction of CMC;
> - `Sect. 5.3`: The second component of DuMCC, CMC, motivated by the findings in `Sect. 5.2`.
>
> The sophisticated logic detailed above may require some pages to clarify potential confusion arising from our novel task and method. Admittedly, our preference for clarity, on the other hand, made us have to relocate some details to the appendix to adhere to page limits, exactly as the reviewer suggested.
>
> In the revision, we will strive to achieve a better balance by incorporating as many details as possible within the scope of the additional content page allowed. Our detailed plan is as follows:
> - Relocating portions of Sect. B and Sect. C to Sect. 6;
> - Reorganising the architectural details in Sect. D into narrative texts and transferring the associated experimental details from Sect. D to Sect. 6.
>
>
> ---
>
> `W2.` **Limited connection between the main paper and the appendix**
>
> >"The authors vaguely direct readers to Section D for further details, without specifying specific tables or paragraphs, which makes it hard to locate relevant information."
>
> `Response:` We apologise for any inconvenience caused by the insufficient linkage between the main paper and the appendix, which may have extended the time required for reviewing our paper. Following the transfer of experimental details from the appendix to the main paper in `W1`, we will endeavour to enhance the connectivity of the remaining implementation details within the appendix to the main paper in our revision.
>
> ---
>
> `W3.` **Vague heterogeneous GRAMA descriptions**
>
> >"The explanation regarding the heterogeneous GRAMA in Line 123 actually makes me confused. The statement is too vague. What exactly are the key differences between the homogeneous and heterogeneous cases of GRAMA?"
>
> `Response:` We apologise for not having made our heterogeneous GRAMA clearer, which may have confused the reviewer. We would like to clarify that the heterogeneous case of GRAMA serves as a complementary extension to the proposed homogeneous GRAMA framework in our paper, where pre-trained parent models share similar architectures and purposes. We will explicitly highlight and detail this distinction in Sect. 3 of our revised version.
>
> ---
>
> `W4.` **Potential solutions for heterogeneous GRAMA**
>
> >"The authors should at least provide some insights or potential solutions for solving heterogeneous GRAMA, or perhaps consider removing this statement of heterogeneous GRAMA since this is not the contribution of this paper."
>
> `Response:` We appreciate the advice from the reviewer. A potential solution could involve Partial GRAMA, which entails first identifying shared features between two parent models that have different architectures or handle varied tasks. Subsequently, this possible method would yield a multi-head child GNN that selectively integrates elements of the pre-trained parent GNNs. Although this method would increase the model size compared to full GRAMA, which combines the entire models, it would be expected to offer a more favourable balance and trade-off between model size and performance. We will incorporate this discussion in the revised paper.
>
> ---
>
> `W5.` **Applying GRAMA to variants of GNNs**
>
> >"I think it is at least worth a try to apply GRAMA to two different GNN variants, such as GCN and GraphSage, since the key learnable parameters in these GNNs are all MLPs."
>
> `Response:` The reviewer's point is very well taken. Our immediate-next goal is, exactly as the reviewer suggested, to adapt our proposed method for cross-architecture model reuse scenarios, including the combination of GCN and GraphSAGE as outlined in lines 348-349 of our paper.
>
> Here, to echo the reviewer's concern, we conducted a pilot study by performing additional experiments on GRAMA using both a GCN and a GraphSAGE model on non-overlapping partitions of the ogbn-arxiv dataset. These two parent models share a similar overall architecture but differ in their respective GCN and GraphSAGE layers. For the GraphSAGE model, we employed the official SAGEConv implementation from the Deep Graph Library (DGL), setting the 'aggregator_type' to 'gcn' for this preliminary study. This adjustment is due to the design of our homogeneous GRAMA, which requires alignment of parameters with the same shape and number. The results, as shown below, highlight the promising potential and feasibility of cross-architecture GRAMA applications.
>
> | Parent Models | Architectures | Performance: Dataset 1 | Performance: Dataset 2 |
> |:----- | :----- | :----- | :----- |
> | Parent 1 | GCN | 0.6908 | 0.5752 |
> | Parent 2 | GraphSAGE | 0.6181 | 0.7508 |
> | Child | GCN | 0.6082 | 0.5746 |

---

> > ### Comment · Reviewer_Uv5e · 2024-08-08
> > **Quesiton about homogeneous GRAMA**
> >
> > I want to thank the authors for their detailed responses and the extensive efforts put into conducting new experiments. Most of my questions have been answered satisfactorily. However, I still find the clarification on heterogeneous GRAMA somewhat vague. As I mentioned in my review, in the case of two pre-trained GNN with exactly the same architecture but trained for different domain tasks, should this scenario be classified as homogeneous or heterogeneous GRAMA? I would appreciate further elaboration on this point.

---

> > > ### Author Response · Authors · 2024-08-09
> > > **Response to Reviewer Uv5e**
> > >
> > > >"I want to thank the authors for their detailed responses and the extensive efforts put into conducting new experiments. Most of my questions have been answered satisfactorily. However, I still find the clarification on heterogeneous GRAMA somewhat vague. As I mentioned in my review, in the case of two pre-trained GNN with exactly the same architecture but trained for different domain tasks, should this scenario be classified as homogeneous or heterogeneous GRAMA? I would appreciate further elaboration on this point."
> > >
> > > `Response:` We appreciate the reviewer's follow-up discussion and would like to apologise for not explicitly defining and explaining heterogeneous GRAMA in our paper and the rebuttal.
> > >
> > > We would like to clarify that heterogeneous GRAMA refers to scenarios where pre-trained parent models have either diverse architectures or are designed for different domain tasks. Indeed, the case mentioned by the reviewer falls under heterogeneous GRAMA.
> > >
> > > In our revision, we will address the reviewer's comments by explicitly defining heterogeneous GRAMA. Specifically, we will replace the sentence in lines 123-124 of the main paper, "The exploration into varied architectures and tasks (i.e., heterogeneous GRAMA) remains a topic for subsequent future studies", with the following:
> > >
> > > "We reserve the exploration of more challenging heterogeneous GRAMA scenarios, where pre-trained parent models either have diverse architectures or are designed for different domain tasks, as a topic for future studies."

---

> > > > ### Comment · Reviewer_Uv5e · 2024-08-11
> > > > **Final rating**
> > > >
> > > > I thank the authors for sending their detailed further clarification. I've updated my rating accordingly, considering that the responses have resolved all of my concerns satisfactorily. I am also pleased that the authors are open to further discussions. I hope the authors attend to the main points raised in the reviews as they prepare an updated paper version.

---

> ### Author Response · Authors · 2024-08-06
> **Response to Reviewer Uv5e (Part 2/2)**
>
> `W6.` **Comparative upper bound results of re-training**
>
> >"In Tables 2 to 5, the authors overlook an important comparative result: the performance of retraining a multi-dataset model that can jointly combine the expertise of both parent models."
>
> `Response:` We appreciate the reviewer's constructive suggestion. Following the reviewer's advice, we conducted additional experiments by re-training a model using combined parent datasets with ground-truth labels. The results are presented below.
>
> As the reviewer pointed out, these results can indeed establish an upper bound for GRAMA's performance. We will incorporate these results into Sect. 6 and provide the corresponding discussion in the revision.
>
> | Tables | Datasets  | Parent 1 | Parent 2 | Re-training |
> |:----- | :----- | :----- | :----- | :----- |
> | Tab. 2 | ogbn-arxiv | 0.7193 / 0.5516 |  0.6564 / 0.7464 | 0.6903 / 0.7268 |
> | Tab. 2 | ogbn-products | 0.7982 / 0.7308 | 0.7626 / 0.7904 | 0.7981 / 0.7787 |
> | Tab. 3 | ogbg-proteins | 0.7478 | 0.7222 | 0.7514 |
> | Tab. 3 | ogbn-molbace, ogbg-molbbbp | 0.7247 / 0.4681 | 0.4067 / 0.6366 | 0.6954 / 0.6087 |
> | Tab. 4 | ModelNet40 | 0.9159 / 0.8151 | 0.8862 / 0.9275 | 0.9390 / 0.9243 |
> | Tab. 5 | S3DIS | 0.8181 | 0.8174 | 0.8428 |
>
> ---
>
> `W7.` **Experiments with additional splitting ratios**
>
> >"It is also not very clear to me why the authors chose a 20%/80% splitting ratio in the experiments, as there are too few explanations about the splitting protocol. For example, is the split random? Would other splitting, such as 10/90, work for GRAMA?"
>
> `Response:` We would like to thank the reviewer for the constructive comments. In fact, we have indeed addressed the choice of our dataset partition strategy in lines 299-302 of the main paper, following the well-established methodology of "Git Re-basin" by Ainsworth et al. [1], which involves random partitioning of the dataset.
>
> Following the reviewer's suggestion, we conducted further experiments with an additional 90%/10% random split on the relatively large-scale ModelNet40 dataset, used for 3D object recognition tasks. The results, presented below, further substantiate the effectiveness of our proposed method.
>
>
> Datasets | Parent 1 | Parent 2 | Ours (w/o CMC) | Ours (w/ CMC)
> |:----- |:----- | :----- | :----- | :----- |
> | Dataset 1 | 0.9246 | 0.8393 |  0.8451 | 0.8782 |
> | Dataset 2 | 0.7621 | 0.9294 |  0.8374 | 0.8422 |
>
>
> ---
>
> `W8.` **Constrained limitation discussions**
>
> >"The current discussion of limitations in this paper appears somewhat constrained. The authors should consider expanding this section, either in the main body of the paper or in the appendix. An additional limitation to consider would be the scenario where two pre-trained models address tasks at different levels, such as node and graph-level tasks."
>
> `Response:` The reviewer's point is well taken. Indeed, our current DuMCC framework does not accommodate scenarios where the parent models tackle tasks at different levels, a challenge that falls within the scope of heterogeneous GRAMA as outlined in `W3`. To address this, we will enhance our discussion on limitations by introducing a new Sect. G titled "Limitation Discussion" in the appendix. This section will offer a detailed analysis of this limitation and provide the corresponding potential solutions for such cases, as exemplified in `W4`.
>
> ---
>
> `W9.` **Typos**
> >"Line 26: 'reduing' - 'reducing';
> Line 106: 'presents' - 'present';
> Line 193: 'are' - 'is'."
>
> `Response:` We appreciate the reviewer's thorough examination of our paper. We will address these typographical errors in our revisions by correcting "reduing" to "reducing" in line 26, removing the extraneous "s" from "presents" in line 106, and adjusting "are" to "is" in line 193 to ensure grammatical correctness.

---

### Official Review · Reviewer_856w · 2024-07-11

**Soundness:** 3
**Presentation:** 2
**Contribution:** 3
**Rating:** 6
**Confidence:** 4

**Summary:**

The paper tackles pre-trained model fusion for graph-centric tasks. The pre-trained models share the same architecture, but differ in the graph datasets they are trained on. The pipeline involves two core approaches. The first one matches parameters in pre-trained parent models by aligning the aggregated messages of the pre-trained parent models. The second one modifies the message statistics for the child model to correspond with the overall statistics of the pre-trained models. The authors have validated two approaches on diverse datasets and models and show nice visualization results.

**Strengths:**

S1. As far as I know, this is the first work to study model fusion for graph tasks. The idea is novel and has potential applicability to other networks like Transformers.

S2. Throughout the paper, the author examines alternative approach and the challenges associated with each to more effectively justify their choices.

S3. The authors show quantitative and qualitative results on diverse datasets for convincing validation throughout their experiments. I found the visualization results in the experiments to be nice and show the effectiveness of the approach.

**Weaknesses:**

W1. Discussions on existing model fusion approaches for CNNs are not adequate. The presentation will benefit from a more comprehensive literature review.

W2. Lack of sensitivity analysis on the interpolation coefficient alpha.

W3. It seems that each task uses only one network architecture and dataset partition. More experiments are needed with more network architectures.

W4. It will be great to showcase more applications of the proposed approach besides training-free model reuse.

**Questions:**

Address weaknesses as mentioned above.

**Limitations:**

Yes. There are no obvious negative society impacts I can notice.

---

> ### Author Rebuttal · Authors · 2024-08-06
>
> ### **Response to Reviewer 856w**
>
> We would like to thank the reviewer for the very helpful feedback. We will address each of the reviewer's comments in detail as follows.
>
> `W1.` **Inadequate literature review**
>
> >"Discussions on existing model fusion approaches for CNNs are not adequate. The presentation will benefit from a more comprehensive literature review."
>
> `Response:` We appreciate the reviewer for the advice. We will elaborate on the *Model Merging* section in Sect. 2 of the main paper's related work, and add a new section in Sect. E (extended related work) of the appendix to offer a more comprehensive discussion on existing model fusion techniques for CNNs and transformers. Specifically, we plan to broaden the discussions on model fusion for CNNs and transformers in the revision, covering, but not limited to, the following:
> - Singh and Jaggi [r1] frame the task of one-shot neural network merging based purely on model weights as model fusion. They propose an Optimal Transport Fusion (OTFusion) method that uses the Wasserstein distance to align weight matrices prior to performing parameter fusion, eliminating the need for re-training;
> - The subsequent work [r2] extends the application of OTFusion to Transformer-based architectures, aiming to enhance efficiency and performance through fusion;
> - Liu et al. [r3] approach the challenging task of model fusion as a graph matching problem, incorporating second-order parameter similarities to improve the effectiveness of model fusion;
> - Addressing the fusion of pre-trained models trained on disparate tasks, Stoica et al. [r4] develop ZipIt!, a novel method that utilises a "zip" operation for layer-wise fusion based on feature redundancy, creating a versatile multi-task model without additional training;
> - More recently, Xu et al. [r5] present the Merging under Dual-Space Constraints (MuDSC) framework. This approach optimises permutation matrices by mitigating inconsistencies in unit matching across both weight and activation spaces, targeting effective model fusion in multi-task scenarios.
>
> [r1] Singh and Jaggi. Model fusion via optimal transport. In NeurIPS, 2020.
>
> [r2] Imfeld, et al. Transformer fusion with optimal transport. In ICLR, 2024.
>
> [r3] Liu, et al. Deep neural network fusion via graph matching with applications to model ensemble and federated learning. In ICML, 2022.
>
> [r4] Stoica, et al. Zipit! merging models from different tasks without training. In ICLR, 2024.
>
> [r5] Xu, et al. Training-free pretrained model merging. In CVPR, 2024.
>
>
> ---
>
> `W2.` **Sensitivity analysis on $\alpha$**
>
> >"Lack of sensitivity analysis on the interpolation coefficient alpha."
>
> `Response:` We appreciate the reviewer's comments. In fact, the results of sensitivity analysis and the corresponding detailed discussions have already been reported in Sect. C of the appendix. We have also referenced these results of sensitivity analysis in line 292-293 of the main paper. We will further highlight these results in the revision.
>
>
> ---
>
> `W3.` **Experiments with more network architectures**
>
> >"It seems that each task uses only one network architecture and dataset partition. More experiments are needed with more network architectures."
>
> `Response:` We appreciate the constructive feedback from the reviewer. Following the suggestions provided, we conducted additional experiments with various network architectures and dataset splits on the large-scale ModelNet40 dataset. The results, along with detailed descriptions of the architectures used, are shown below. We will incorporate these results into the revised version of the paper.
>
>
> | Architectures | Layers | Feature Map Channels | MLP |
> |:----- |:----- | :----- | :----- |
> | Architecture-Main |   8   | [64, 64, 128, 256, 1024] | [512, 256, 40] |
> | Architecture-Rebuttal |   7   | [32, 32, 64, 128, 512] | [256, 40] |
>
>
> | Architectures | Partition | Datasets | Parent 1 | Parent 2 | Ours (w/o CMC) | Ours (w/ CMC)
> |:----- |:----- |:----- | :----- | :----- | :----- | :----- |
> | Architecture-Main | 10%/90% | Dataset 1 | 0.9246 | 0.8393 | 0.8451 | 0.8782 |
> | Architecture-Main | 10%/90% | Dataset 2 | 0.7621 | 0.9294 | 0.8374 | 0.8422 |
>
>
> | Architectures | Partition | Datasets | Parent 1 | Parent 2 | Ours (w/o CMC) | Ours (w/ CMC)
> |:----- |:----- |:----- | :----- | :----- | :----- | :----- |
> | Architecture-Rebuttal | 20%/80% | Dataset 1 | 0.9184 | 0.8920 | 0.8236 | 0.8846 |
> | Architecture-Rebuttal | 20%/80% | Dataset 2 | 0.8175 | 0.9299 | 0.8279 | 0.8550 |
>
>
> ---
>
> `W4.` **Additional applications**
>
> >"It will be great to showcase more applications of the proposed approach besides training-free model reuse."
>
> `Response:` We would like to thank the reviewer for the constructive comments. Here, we demonstrate an additional application of the proposed DuMCC beyond its primary role in training-free model reuse, specifically as a pre-processing step that facilitates more effective graph-based knowledge amalgamation (KA). In the table below, "KA" refers to re-training a student model from random initialisations by amalgamating knowledge from two teachers pre-trained on distinct partitions of the ModelNet40 dataset. "KA + GRAMA" denotes the process of performing KA by fine-tuning from the child model obtained via the proposed DuMCC method. The results indicate that "KA + GRAMA" leads to improved knowledge amalgamation performance. Additionally, since GRAMA provides a superior initialisation compared to random initialisation, "KA + GRAMA" empirically achieves faster convergence speed.
>
>
> | Methods | Performance: Dataset 1 | Performance: Dataset 2 | Performance: Average |
> |:----- | :----- | :----- | :----- |
> | KA | 0.9246 | 0.9270 | 0.9258 |
> | KA + GRAMA | 0.9221 | 0.9318 | 0.9269 |

---

> > ### Comment · Reviewer_856w · 2024-08-10
> > **Response to the rebuttal**
> >
> > Thanks for the well-structured rebuttal. I'm happy with the authors' reply and particularly appreciate the well-prepared and ready-to-incorporate text in the rebuttal, which strongly convinces me that the authors will make the corresponding necessary changes to improve the manuscript.
> >
> > Generally speaking, there is no doubt that the paper makes a clear contribution to both the fields of model fusion and gnn, with novel insights like the amplified parameters sensitivities due to topology, which adds new knowledge to the field. After careful consideration for my final justification, I have decided to raise my score to 6: weak acceptance.
> >
> > There is one minor issue remaining. When reviewing again the related work on model fusion, I noticed several works in the reference list have the wrong pubilcation year. For instance, the well-known git rebasin work ([1] in the paper) was actually published at last year's ICLR, NOT 2022; Zipit [57] was at this year's ICLR, not in 2023. Double check the reference list for such errors.

---

> > > ### Author Response · Authors · 2024-08-11
> > > **Response to Reviewer 856w**
> > >
> > > >"There is one minor issue remaining. When reviewing again the related work on model fusion, I noticed several works in the reference list have the wrong pubilcation year. For instance, the well-known git rebasin work ([1] in the paper) was actually published at last year's ICLR, NOT 2022; Zipit [57] was at this year's ICLR, not in 2023. Double check the reference list for such errors."
> > >
> > > `Response:` We sincerely appreciate the reviewer for the positive support and the constructive comment regarding the issues in our reference entries. We have thoroughly reviewed all the entries in the reference section and will correct the entries accordingly in our revision as follows:
> > >
> > > *Original references:*
> > >
> > > "
> > >
> > > [1] Samuel Ainsworth, Jonathan Hayase, and Siddhartha Srinivasa. Git re-basin: Merging models modulo permutation symmetries. In ICLR, 2022.
> > >
> > > [26] Moritz Imfeld, Jacopo Graldi, Marco Giordano, Thomas Hofmann, Sotiris Anagnostidis, and Sidak Pal Singh. Transformer fusion with optimal transport. In ICLR, 2023.
> > >
> > > [57] George Stoica, Daniel Bolya, Jakob Brandt Bjorner, Pratik Ramesh, Taylor Hearn, and Judy
> > > Hoffman. Zipit! merging models from different tasks without training. In ICLR, 2023.
> > >
> > > [61] Hongyi Wang, Mikhail Yurochkin, Yuekai Sun, Dimitris Papailiopoulos, and Yasaman Khazaeni. Federated learning with matched averaging. In ICLR, 2019.
> > >
> > > "
> > >
> > > *Revised references:*
> > >
> > > "
> > >
> > > [1] Samuel Ainsworth, Jonathan Hayase, and Siddhartha Srinivasa. Git re-basin: Merging models modulo permutation symmetries. In ICLR, 2023.
> > >
> > > [26] Moritz Imfeld, Jacopo Graldi, Marco Giordano, Thomas Hofmann, Sotiris Anagnostidis, and Sidak Pal Singh. Transformer fusion with optimal transport. In ICLR, 2024.
> > >
> > > [57] George Stoica, Daniel Bolya, Jakob Brandt Bjorner, Pratik Ramesh, Taylor Hearn, and Judy
> > > Hoffman. Zipit! merging models from different tasks without training. In ICLR, 2024.
> > >
> > > [61] Hongyi Wang, Mikhail Yurochkin, Yuekai Sun, Dimitris Papailiopoulos, and Yasaman Khazaeni. Federated learning with matched averaging. In ICLR, 2020.
> > >
> > > "

---

### Official Review · Reviewer_rPnJ · 2024-07-23

**Soundness:** 3
**Presentation:** 3
**Contribution:** 3
**Rating:** 6
**Confidence:** 5

**Summary:**

The paper is interested in re-using Graph Neural Networks trained from one task to another task (e.g., transfer learning). This motivation is nice. We've seen such trends in computer vision (e.g., network trained on ImageNet used for CIFAR), or more recently, everyone is using LLMs for a variety of tasks they were not trained on (by prompt engineering, LoRA, etc). The paper proposes to do the same for graphs. Their specific methodology starts by combining multiple GNNs. To do so, they want to permute the internal channels of the graph neural network so that the networks are best-aligned, before combing them.

**Strengths:**

* Using Pre-trained networks can save compute resources
* If the initial trained dataset is so large and/or private, it might not be practical to retrain the network (even with much compute resources). Using pre-trained networks would work well here.
* Paper proposes to use multiple GNN (pre-trained) networks for a new task. The method is simple: just average the latent vectors of the GNNs. However, for this averaging to make sense, authors propose permuting the channels of the GNN latent vectors. This permutation is optimized using an objective (and algorithm).

**Weaknesses:**

While the paper is well-motivated and well-written, it nicely builds-up the reader's excitment to anticipate for the actual method. Once reader arrives to the well-anticipated Equation (3), reader discovers that the equation is not properly-written ("buggy"). I've written many papers with probably subtle bugs in them (e.g., perhaps some footnote or some text in some side-section has a bug). However, a bug in the main equation qualifies for an immediate rejection.

## Math incorrectness
After Eq 1:

"where $\mathbf{P}^∗$ represents the permutation matrices" -- what does that mean? The expression $\mathbf{P}^∗$  is not used in the Equation. I **think** it can imply one of two things. Option 1: $\mathbf{P}^∗ = [P^{(\ell)} ]_{\ell \in [L]}$; or Option 2: $\mathbf{P}^∗$ refers to the space of permutation matrices (e.g., a reshuffle of the identity matrix or its continuous relaxation which is probably any basis?)

Equation 3 has several issues:

1. Where does $i$ come from? do you mean to add $\sum_i$ between the $\arg\min$ and the Frobenius norm?
2. The aggregation function "Agg" is not introduced. While the exact definition is unnecessary, however, the domain and range should be clearly specified. Does it take a many vectors and output one vector? If yes, then $P \cdot \textrm{Agg}$ should be a vector and therefore you should be minimizing the L2 norm (not Frobenius norm). If it acts on the whole graph (i.e., $\textrm{Agg}$ outputs a matrix with shape `NumNodes x FeatureDim`), then you are multiplying the permutation matrix from the wrong side.
3. In general, where do the $X$s come from? I was expecting data-independent scheme. In my reading, at this point, I think paper **is "learning** to align" (because you are doing gradient decent on the permutation matrix by looping over data, no?)

-- with all honesty, I stopped reading the paper after arriving at Eq3 and skimming over the algorithm. Please address the above and I am happy to take another look during the rebuttal. That being said, I do trust that your implementation is correct, but the paper may not indeed reflect the implementation.

**Questions:**

- Does the exact form of Equation 2 depends on the architecture of the GNN model? I would expect to see different formulas depending on the model (e.g., GraphSAGE vs GCN of Kipf vs GAT vs MixHop etc)

- Please respond to the main math weakness, which might cause you to be explicit about the notation

**Limitations:**

I did not spot a Limitations section.

---

> ### Author Rebuttal · Authors · 2024-08-06
>
> ### **Response to Reviewer rPnJ (Part 1/2)**
>
> We appreciate the reviewer's constructive comments and thoughtful suggestions, and we sincerely apologise for any confusion or ambiguity caused by our use of notations.
>
> We are committed to addressing each of the reviewer's concerns as outlined below. Due to character constraints, we have to divide our responses into two parts. The second part will be presented as a comment appended to our initial rebuttal.
>
> Additionally, we warmly welcome any further questions or comments the reviewer may wish to share.
>
>
> `W1.` **Clarity on $\mathbf{P}$***
>
> >"After Eq 1: 'where $\mathbf{P}^*$ represents the permutation matrices' -- what does that mean? The expression $\mathbf{P}^*$ is not used in the Equation. I *think* it can imply one of two things. Option 1: $\mathbf{P}^* = \left[\mathbf{P}^{(\ell)}\right]_{\ell \in [L]}$ or Option 2: $\mathbf{P}^*$ refers to the space of permutation matrices (e.g., a reshuffle of the identity matrix or its continuous relaxation which is probably any basis?)
>
> `Response:` We apologise for not having clearly defined the symbol $\mathbf{P}^*$, which may have led to confusion and inadvertently extended the review process. We appreciate the reviewer's patience and constructive comments. We would like to clarify that $\mathbf{P}^*$ denotes the set of all permutation matrices corresponding to each layer $\ell$ of the graph neural network (GNN), which is exactly Option 1 suggested by the reviewer.
>
> We would like to further clarify that while $\mathbf{P}^*$ does not appear in Eq. 1, it is utilised in line 158, Eq. 3, Conjecture 4.1, and lines 208, 210, and 222. We intended to use this notation to simplify the discussions related to the collection of permutation matrices across different layers.
>
> Admittedly, we sincerely apologise for the oversight in not explicitly linking Eq. 1 with $\mathbf{P}^*$ and for not adequately defining and elaborating on this symbol, which resulted in ambiguity. To address this issue, we will revise Eq. 1 in line with the reviewer's constructive suggestion and enhance the description of $\mathbf{P}^*$ in our revision by incorporating the following:
>
> "$$
> W^{(\ell)} = \alpha W_{a}^{(\ell)} + (1 - \alpha) P^{(\ell)} W_{b}^{(\ell)} (P^{(\ell-1)})^{T}, \quad P^{(\ell)} \in \mathbf{P}^*, \tag{1}
> $$
> where $\mathbf{P}^* = \left[{P}^{(\ell)}\right]_{\ell \in [L]}$ represents the set of all permutation matrices $P^{(\ell)}$ for each layer $\ell$ of the graph neural network (GNN). Here, $[L]$ refers to the set of indices corresponding to all layers in the GNN."
>
> ---
>
> `W2.` **Issues on Eq. 3**
>
> `W2.1. & W2.2.` *Clarity on $i$ and "Agg"*
>
> >"Where does $i$ come from? do you mean to add $\sum_i$ between the $\arg \min$ and the Frobenius norm?"
>
> >"The aggregation function 'Agg' is not introduced. While the exact definition is unnecessary, however, the domain and range should be clearly specified. Does it take many vectors and output one vector? If yes, then $P \cdot \text{Agg}$ should be a vector and therefore you should be minimizing the L2 norm (not Frobenius norm). If it acts on the whole graph (i.e., Agg outputs a matrix with shape NumNodes $\times$ FeatureDim), then you are multiplying the permutation matrix from the wrong side."
>
> `Response:` We sincerely appreciate the reviewer's constructive feedback and apologise once again for the lack of clarity in our notations and the rigor of our equations. Since the issues concerning $i$ and $\text{Agg}$ are interrelated, we are combining our responses to W2.1 and W2.2 to address these two issues collectively.
>
> Originally, our intention was to use the symbol $i$ to universally represent the set of all nodes, rather than a single node. Consequently, $\text{Agg}$ was meant to act on the entire graph, outputting a matrix. Here, we would like to clarify that this output matrix was shaped as FeatureDim $\times$ NumNodes. This configuration was deliberately chosen to align with the conventions used in model merging within the Euclidean domain (e.g., Eq. 1 in [r1] and Eq. 13 in [r2], where the corresponding matrices are denoted as Dim $\times$ Num). This alignment was intended to facilitate a unified formulation and simplify conceptually intuitive comparisons across different domains. As a result, in our original Eq. 3, we multiplied the permutation matrix from the left side to match this dimension. This choice also explains our use of the Frobenius norm, as the operation is performed on the matrix.
>
> Admittedly, exactly as the reviewer kindly suggested, the definition and use of the symbol $i$ in this context lacks mathematical rigor, correctness, and explicitness, which subsequently led to confusion concerning $\text{Agg}$ and the Frobenius norm. We sincerely appreciate the reviewer's thorough advice on this issue. In our revision, we will address this issue by exactly following the reviewer's suggestion: defining $i$ as a single node identifier and introducing $\sum_i$ after $\arg \min$ to explicitly iterate over all nodes. This adjustment will result in the output of $\text{Agg}$ being a vector. Accordingly, we will replace the original Frobenius norm with the L2 norm to ensure consistency with these changes.
>
> We sincerely thank the reviewer once again for the insightful comments. We will revise Eq. 3 as described above, along with including the corresponding detailed descriptions for rigorous and explicit formulation and symbol definitions. Additionally, we will also update Eq. 3 in Alg. 1 accordingly.
>
>
> [r1] Ainsworth, et al. Git re-basin: Merging models modulo permutation symmetries. In ICLR, 2023.
>
> [r2] Li, et al. Deep model fusion: A survey. 2023.

---

> ### Author Response · Authors · 2024-08-07
> **Response to Reviewer rPnJ (Part 2/2)**
>
> `W2.3(a)` *Clarity on $\mathbf{X}$s*
>
> >"In general, where do the $\mathbf{X}$s come from? I was expecting data-independent scheme."
>
> `Response:` The reviewer's point is very well taken. In our paper, we initially introduced a fully data-independent approach for GRAMA, referred to as the vanilla VAPI method, described in lines 158-161: "One possible data-independent solution is to minimise the L2 distance between the weight vectors of the pre-trained models by solving a sum of bilinear assignments problem, similar to weight matching techniques described in [1]."
>
> However, as discussed in Sect. 4.2, our subsequent analysis reveals that GNNs are particularly sensitive to mismatches in parameter alignment, which are "contingent upon the topological characteristics inherent to each graph" (Conjecture 4.1). This sensitivity renders the data-independent solution less effective. Motivated by this, we propose integrating the topological characteristics inherent to each graph. To achieve this, we have to resort to passing the graph data to the pre-trained model to capture these graph-specific topological characteristics, utilising $\mathbf{X}$.
>
> Nevertheless, we clarify that our PMC and CMC methodologies require only a single forward pass of the unlabelled graph data to extract messages for alignment and calibration, respectively—eliminating the need for iterative training or ground-truth labels. Our immediate-next goal is, exactly as the reviewer suggested, to explore the possibility of an entirely data-independent GRAMA scheme, for example, by initially generating fake graphs as described in [r3]. We will include these discussions in the revised version.
>
>
> [r3] Deng and Zhang. Graph-free knowledge distillation for graph neural networks. In IJCAI, 2021
>
>
> `W2.3(b)` *Details of aligning*
> >" In my reading, at this point, I think paper is 'learning to align' (because you are doing gradient descent on the permutation matrix by looping over data, no?)"
>
> `Response:` We apologise for any confusion caused by our previous lack of clarity and the omission of essential details. We would like to clarify that our method does not employ gradient descent for alignment. In the field of model merging within the Euclidean domain, the minimisation problem in the form of Eq. 3 is typically transformed into a maximisation problem to maximise an inner product (as derived from expanding Eq. 3), thereby fitting it within the framework of a standard linear assignment problem [r1, r2, r3, r4]. In our revision, we will include these crucial details and introduce a preliminary section on Euclidean model merging to ensure our paper is self-contained.
>
>
> [r1] Ainsworth, et al. Git re-basin: Merging models modulo permutation symmetries. In ICLR, 2023.
>
> [r2] Li, et al. Deep model fusion: A survey. 2023.
>
> [r3] Liu, et al. Deep neural network fusion via graph matching with applications to model ensemble and federated learning. In ICML, 2022.
>
> [r4] Stoica, et al. Zipit! merging models from different tasks without training. In ICLR, 2024.
>
>
> ---
>
> `Q1.` **Model-specific variants of Eq. 2**
>
> >"Does the exact form of Equation 2 depends on the architecture of the GNN model? I would expect to see different formulas depending on the model (e.g., GraphSAGE vs GCN of Kipf vs GAT vs MixHop etc)"
>
> `Response:` We appreciate the constructive comments provided by the reviewer. To maintain clarity and simplicity, Eq. 2 in our paper is based on the most basic form of GNNs. Due to the short rebuttal period, we have only been able to develop specific formulas for GraphSAGE and Kipf's GCN as suggested by the reviewer. Our model formulation presented here follows the unified mathematical framework detailed in [r5], albeit with symbols adapted to those used in our paper. In the revised version, we are committed to developing more detailed, model-specific formulas.
>
> (We apologise for having to split each of our equations below into two lines for display purposes, as OpenReview does not support single-line presentation of long equations.)
>
> *Eq. 2 tailored for GCN of Kipf is as follows:*
> $$
> \Delta F_i \approx \sigma'\left(W  \sum_{j \in \mathcal{N}(i)} \frac{X_j}{\sqrt{\deg}_i \sqrt{\deg}_j}\right)
> $$
>
> $$
> \cdot \left(\epsilon  \sum_{j \in \mathcal{N}(i)} \frac{X_j}{\sqrt{\deg}_i \sqrt{\deg}_j}\right)
> $$
>
> *Eq. 2 tailored for GraphSAGE is as follows:*
> $$
> \Delta F_i \approx \sigma' \left(W \ \text{Concat}\left(X_i, \text{Mean}_{j \in \mathcal{N}(i)} X_j\right)\right)
> $$
>
> $$
> \cdot \left(\epsilon \ \text{Concat}\left(X_i, \text{Mean}_{j \in \mathcal{N}(i)} X_j\right) \right)
> $$
>
>
> [r5] Dwivedi, et al. Benchmarking graph neural networks. In JMLR, 2023.
>
>
> ---
>
> `L1.` **Limitations**
>
> >"I did not spot a Limitations section."
>
> We apologise for not making the limitations section more explicit in our paper. In fact, we have included a section on limitations in Sect. 7, titled "Conclusions and Limitations". We will highlight this section further in the introduction.

---

> > ### Comment · Reviewer_rPnJ · 2024-08-12
> >
> > Since the reviewers have responded to all my comments, I am increasing my score.
> >
> > Thank you for your good work!

---

### Decision · Program_Chairs · 2024-09-25

**Decision:**

Accept (poster)

**Comment:**

The aim of the paper is the reuse of GNN models for transfer to new tasks without re-training, fine-tuning or annotated labels. It proposes a new Dual-Message Coordination and Calibration methodology to solve the problems of vanilla permutation-based approaches.

The reviewer consensus is that the paper is well motivated addressing an interesting and important problem in a novel way. It contributes to both the model fusion and GNN domains. The method is well presented and the experiments document convincingly its good performance.

The submitted version of the paper raised numerous questions and requests for clarifications or improvements. The authors addressed most of these in their extensive and meticulous rebuttal to the satisfaction of the reviewers. As it was these additional pieces of information - e.g. new experimental results, corrected notation, improved explanations - that convinced the reviewers about the quality of the contribution, we urge the authors to update the final version of the paper accordingly.